# Comparative Proteomics of Two Flor Yeasts in Sparkling Wine Fermentation: First Approach

**DOI:** 10.3390/foods14020282

**Published:** 2025-01-16

**Authors:** Juan C. García-García, Teresa García-Martínez, Juan J. Román-Camacho, Juan Moreno, Juan C. Mauricio

**Affiliations:** Department of Agricultural Chemistry, Edaphology and Microbiology, Agrifood Campus of International Excellence ceiA3, University of Córdoba, 14014 Córdoba, Spain; p22gagaj@uco.es (J.C.G.-G.); b32rocaj@uco.es (J.J.R.-C.); qe1movij@uco.es (J.M.); mi1gamaj@uco.es (J.C.M.)

**Keywords:** *Saccharomyces cerevisiae*, proteins, sparkling wine, second fermentation

## Abstract

The traditional method is considered the highest-quality sparkling wine making technique. Its main characteristic is that the entire sparkling transformation takes place in the bottle, producing complex, refined wines with fine, persistent bubbles. Currently, the second fermentation in the bottle is initiated by a few commercially available strains of *Saccharomyces cerevisiae*. This lack of yeast diversity leads to a predominant uniformity in the sensory profiles of the final products and a lack of distinctive wines. The aim of the present study is to compare the proteomic profiles of the first flor yeast strain (G1) on the market for the production of high-quality sparkling wines with a new flor strain (N62) selected for its specific characteristics for potential use in sparkling wine production, such as flocculation, tolerance to high ethanol concentrations, and β-Glucosidase-positivity, which is valuable for improving wine aroma complexity. The results showed that these strains behaved differently in the middle fermentation tested: the strain that reached 3 atmospheres faster was strain N62, which achieved higher growth, viability, glycerol content, and volatile acidity. In G1, a higher ethanol content was reached, and lower growth and viability were observed. Key protein data support the relationship between these differences, and the proteomic analysis could show that strain N62 had a higher abundance of proteins related to protein synthesis, such as PAB1, TEF2, and RPL25; DAK1, GPP1, and GPP2 are involved in glycerol synthesis and PDC6 and ALD4 in acetate synthesis. In the case of G1, the abundance of ADHI is associated with ethanol production and cell wall proteins with YGP1, EXG1, SCW11, PST1, CIS3, and PIR3, while the onset of autophagy is associated with PRC1, PRB1, ATG42/YBR139W, PRE8, PRE9, and PUP2.

## 1. Introduction

Sparkling wines have a high economic value, and their consumption and production have increased significantly in recent years, driven by various trends and changes in the market. Sparkling wines are a unique category of wine characterized by a significant amount of carbon dioxide, which produces its characterized effervescence when the bottle is uncorked. This effervescence results from an overpressure of CO_2_, which can be exogenous in the case of carbonated wines, or endogenous in the case of sparkling wines [1,2,3]. The latter can be produced with one (ancestral method) or two alcoholic fermentations. The second fermentation can be carried out in an isobaric tank (Charmat method) [4] or in a closed bottle (traditional or Champenoise method), which is the most widely used method [5]. This second fermentation takes place in bottles to which base wine and tirage liqueur are added, consisting of sugar (approximately 20–24 g of sucrose/L), yeasts preadapted for the successful second fermentation (1 to 2 × 10^6^ cells/mL), and a riddling agent [6]. Therefore, the starter cultures selected for sparkling wine production must have several technological characteristics in addition to those suggested for yeast strains used in primary fermentation [7]. The use of selected autochthonous yeasts for second fermentation has recently been proposed to enhance the specific features of typical regional wines and to prevent fermentative problems [8,9]. Autochthonous starter cultures represent a potentially important key to wine quality, due to their possible differential adaptation to specific environmental conditions and their prospective contribution to the differentiation of the organoleptic properties of final products [8,9,10,11]. The work of Vigentini et al. [11] testifies the existing interest in the characterization and selection of autochthonous yeasts to improve the diversification of sparkling wines.

Flor yeasts are specialized strains of *Saccharomyces cerevisiae* that form a biofilm on the surface of certain wines, such as sherry, during biological aging. Flor yeasts are innovative and ideal for the sparkling wine industry due to their resistance to high ethanol concentrations, surface adhesion properties that facilitate wine clarification, and the ability to provide a characteristic volatilome and odor profile [12]. Some proteomic studies on wine yeasts have been carried out in the second fermentation in-bottle because this is the level of expression closest to the final product in terms of organoleptic characteristics, and to understand the main biological processes under these conditions [13,14,15]. Previously, we explored the use of a flor yeast strain (*S. cerevisiae* G1) in the second fermentation for sparkling wine production [12]. In this context, the first sparkling wine successfully produced with a flor yeast, named G1, was commercialized by the company Pérez Barquero in collaboration with the University of Córdoba (Spain). In order to increase the diversity and applicability of flor yeast strains in sparkling wine production, we selected a new flor yeast (*S. cerevisiae* N62) and carried out a first proteomic study using the G1 strain as a reference at mid-second fermentation. The main objective is to characterize (through qualitative and quantitative analysis) the proteomic profile of these two yeast strains, which is the closest level to the final product in terms of organoleptic characteristics.

## 2. Materials and Methods

### 2.1. Flor Yeast Strains

Two *S. cerevisiae* yeast strains isolated from wine biofilm (called flor veil) undergoing biological aging at the Pérez Barquero winery in the Montilla-Moriles D.O.P. (Spain) were compared for use in the second fermentation: a G1 strain (ATCC MYA-2451), isolated from a wine with 14.5% *v*/*v* ethanol content and used for the first time in sparkling wine production [12], and an N62 strain (Yeast Collection of the University of Córdoba, Microbiology Department), isolated from wines with 14.0% *v*/*v* ethanol content and selected for its positive β-Glucosidase activity, flocculation, ethanol tolerance, and high CO_2_ production.

### 2.2. β-Glucosidase and Ethanol Tolerance

A total of 100 μL of YPD liquid medium (1% yeast extract, 2% peptone, and 2% glucose) from each strain was inoculated in triplicate onto plates containing β-Glucosidase-detection medium (5 g/L arbutin, 1 g/L yeast extract, 20 g/L agar, and 0.2% of a 1% (*w*/*v*) ferric chloride solution). The plates were incubated at 28 °C for 15 days. Dark black cultures were considered positive. To assess the ethanol tolerance of the selected strains, a series of 15 mL sterile plastic tubes containing 10 mL of a liquid medium with 3 g/L yeast extract, 5 g/L peptone, 10 g/L dextrose, and wine ethanol (*v*/*v*) to give a final ethanol concentration of between 0 and 19% (*v*/*v*) were inoculated with 100 μL of each strain in triplicate. The presence of turbidity, flocculation, or biofilm formation was considered to demonstrate tolerance to ethanol at the given concentration. CO_2_ production was qualitatively assessed by the detachment of the cap during incubation.

### 2.3. Yeast Acclimation and Culture Starter

In the traditional method of sparkling wine production, yeast is acclimatized to stressful base wine conditions in order achieve a successful second alcoholic fermentation. Base wines typically have a low pH, low nutrient concentrations, high acid concentrations, and high ethanol concentrations, and contain sulfur dioxide (SO_2_). In the present study, an acclimation protocol was carried out. Yeast cells were previously grown in YPD medium at 28 °C for 48 h. Later, each strain was inoculated separately (1.5 × 10^6^ cells/mL) in an acclimation medium consisting of 100 mL of 4% sucrose, 0.72 g/L of DAP (diammonium phosphate), and 0.5% of yeast extract and wine alcohol until 12% (*v*/*v*) alcohol was reached *n* consecutive times. These cultures were maintained under gentle constant stirring (100 rpm) on a New Brunswick Scientific (Edison, NJ, USA) orbital shaker at 21 °C for 7 days. Once high cell concentrations (5–6 × 10^6^ cells/mL) and viability (95%) were achieved for both strains, ethanol (12%, *v*/*v*) was added.

### 2.4. Base Wine and Second Fermentation

The base wine was supplied by the company Pérez Barquero (D.O.P. Montilla-Moriles, Córdoba, Spain): Pedro Ximenez white grape variety with glucose and fructose 0.55 g/L; density 0.9994 g/mL; 10.01% *v*/*v* ethanol; pH 3.08; total acidity of 5.30 g/L; free sulfur dioxide 11 mg; total sulfur dioxide 75 mg.

For the second fermentation in-bottle, 750 mL of base wine was added with 24 g/L of sucrose, 0.72 g/L of DAP, and an inoculum of 1.5 × 10^6^ cells/mL of each yeast strain. The bottles were sealed with a stopper and a crown cap. These were placed in a conditioning chamber (12 ± 1 °C and 75% humidity). The second fermentation was monitored by changes in the endogenous pressure of CO_2_, using an internal aphrometer (Oenotilus, Station Oenotechnique de Champagne, Epernay, France). When the pressure reached 3 bars, three independent biological triplicates (bottles) were collected for each yeast; this is the point at which the added sucrose is half consumed and there is a higher protein abundance [13].

### 2.5. Cell Counting: Total and Viable

Total cell number was determined by counting yeast suspensions diluted to 10^−2^ in a Thoma chamber under a binocular optical microscope with the 40× objective lens. Cell viability was determined by growing on YPD agar medium in Petri dishes (1% extract yeast, 2% peptone, 2% dextrose, 2% agar) volumes of 100 µL of the diluted suspension. After an incubation period of 48 h and at the optimal growth temperature of 28 °C, the viable yeast cells grown in this solid medium were enumerated. Counts were made in triplicate.

### 2.6. General Wine Parameters

Ethanol content (%, *v*/*v*), reducing sugars, volatile and total acidity, and pH were quantified according to methods described by OIV [16].

### 2.7. Glycerol Analysis

An enzymatic assay for glycerol was carried out by r-biopharm (Ref. No. E8360, Darmstadt, Germany).

### 2.8. Proteomic Analysis

#### 2.8.1. Cell Collection

The yeast cells from each bottle (sample) were collected by centrifugation for 10 min at 8615× *g*. The samples were washed twice with sterile cold distilled water by homogenization and centrifugation at 4500× *g* for 10 min at 4 °C. Subsequently, the pellet was collected by centrifugation at 16,873× *g* for 1 min at 4 °C.

#### 2.8.2. Cell Lysis and Protein Solubilization

The yeast pellet, previously collected, was resuspended in 500 µL of extraction buffer (100 mM Tris-HCl pH 8.0, 1 mM EDTA, 2 mM DTT and 1 protease cocktail tablet). Next, glass beads (500 µm diameter) were added in a 1:1 ratio with the suspension. Using a mechanical shaking technique, 1 min cycles were applied with a Vibrogen Cell Mill V6 (Edmund Bühler, Bodelshausen, Germany), and for 1 min on ice, for a total of 10 cycles. It was centrifuged at 8615× *g* for 10 min at 4 °C and at 16,873× *g* for 25 min, where cell debris and glass beads were discarded. Protein precipitation was carried out in ice-cold 10% (*w*/*v*) TCA-acetone-DTT with overnight incubation at −20 °C. The samples were then centrifuged at 16,873× *g* for 50 min at 4 °C. After removing the supernatant, the pellet obtained was washed with 500 µL of acetone-DTT (0.07% DTT). The pellet was dried under vacuum in a SpeedVac™ concentrator (Eppendorf 5301, Hamburg, Germany). In total, 500–600 µL of solubilization buffer was added to the pellet. Four 1 min cycles were applied in vortex, staying on ice between each cycle. The protein concentration in the supernatant was estimated by Bradford assay.

### 2.9. LC-MS/MS Analysis

Proteins (100 µg) from each sample were injected for LC-MS/MS analysis at the Central Research Support Service (SCAI), University of Córdoba, Spain. The samples were washed in 1 D SDS-PAGE at 10% polyacrylamide, and 100 V was applied to the electrophoresis resolving gel. A solution of 200 mM ammonium bicarbonate (AB) with 100% acetonitrile was applied for 15 min, followed by 5 min of 50% acetonitrile, to dissolve protein bands for digestion. These were subjected to a reduction process by adding 20 mM AB and incubating for 20 min at 55 °C, followed by alkylation with 40 mM iodoacetamide in 25 mM AB for 20 min in the dark, and washing twice in 25 mM AB. A total of 12.5 ng/µL trypsin (Promega, WI, USA) in 25 mM BA was used for proteolytic digestion. Upon completion of incubation at 37 °C overnight, the digestion process was stopped by adding 1% trifluoroacetic acid, and the digested samples were dried in a SpeedVac™.

Nano LC analyses were carried out on a Dionex Ultimate 3000 nano UHPLC system (Thermo Fisher Scientific, Boston, MA, USA) with an Acclaim Pepmap C18 separation column, 500 mm × 0.075 mm (Thermo Fisher Scientific, Boston, MA, USA). The peptide mixture was trapped using an Acclaim Pepmap C18 precolumn, 5 mm × 0.3 mm (Thermo, Fisher Scientific, MA, USA) at a flow rate of 5 µL/min, at a time of 5 min, with 2% acetonitrile/0.05% trifluoroacetic acid. At 40 °C, the peptides were separated for all runs and eluted with a 150 min gradient of 4–90% acetonitrile/0.1% formic acid solution at a flow rate of 300 nL/min. The extracted peptide cations were transformed into gas-phase ions using an Orbitrap Fusion mass spectrometer (Thermo, Fisher Scientific, MA, USA) equipped with a nanoelectrospray ionization interface. Survey scans of peptide precursors were carried out from 400 to 1500 *m*/*z* at 120 K resolution (at 200 *m*/*z*) with an ion count objective of 4 × 10^5^. Tandem MS was performed using isolation at 1, 2 Th with the quadrupole, CID fragmentation with a normalized collision energy of 35, and fast scan MS analysis in the ion trap. The AGC ion count target was set at 2 × 10^3^, and the maximum injection time was 75 ms. The dynamic exclusion duration was set to 15 s with a tolerance of 10 ppm around the selected precursor and its isotopes, and the selection of monoisotopic precursors was activated. The instrument was run in full speed mode with 3 s cycles, meaning that it would continuously perform MS2 events until the list of non-excluded precursors was reduced to 0 or 3 s.

### 2.10. Protein Identification by Database Search

Raw mass spectrometry data were processed using Proteome Discoverer software (version 2.1.0.81, Thermo Fisher Scientific, Boston, MA, USA). MS/MS spectra were analyzed with the SEQUEST engine against the UniProt database. Searches for peptides resulting from tryptic digestion were performed with the following parameters: up to a missed cleavage, cysteine carbamidomethylation as a fixed modification, and methionine oxidation as a variable modification. Validation of peptide spectral matches (PSMs) was performed with a false discovery rate (FDR) of 1% using a *q*-value-based percolator. Peptide quantification was performed by calculating precursor ion areas with the Precursor Ion Area Detector and normalized using the total peptide quantity mode of Proteome Discoverer. To obtain protein groups, the law of parsimony was applied, and the data were filtered at an FDR of 1%.

### 2.11. Data Analysis

Proteins that were present in at least 50% of the total biological replicates of each sample were retained. The quantitative changes were checked at 3 bars of pressure for each sparkling wine. Protein values were normalized by dividing each by the overall intensity of the sample and then multiplied by the mean global intensity value of all samples. Firstly, proteins obtained in at least 50% of any of the samples were preserved and represented in an intersection diagram. For hierarchical clustering and heat map analysis, proteins identified in 100% of the biological replicates of each sample were used. Mean quantification values were previously scaled and centered using z-score transformation, and then Pearson correlation was applied. One-way ANOVA followed by HSD Tukey test was calculated using the R functions “lm” and “anova”, using the *q*-value to apply multiple testing correction to the *p*-value. Proteins identified in only one biological replicate were removed from the overall count.

The biological function of the protein groups was also studied by constructing protein–protein interaction network maps (INM) using the STRING v12.0 database (accessed on 19 September 2024). A high confidence interaction (score = 0.7) was used, and protein annotations were based on Gene Ontology databases (accessed on 19 September 2024).

## 3. Results

### 3.1. Screening Results for Yeast Strain Selection

Of a total of 119 yeast strains isolated from flor veils of the Pérez Barquero winery, *S. cerevisiae* N62 was selected for its suitability for use in sparkling wine production: 98 strains showed growth around 10% (*v*/*v*) ethanol and N62 above 12% (*v*/*v*) ethanol, β-Glucosidase + (arbutin test positive for 8 of the 119 strains), flocculation + (73 positives of the 119 strains), and cap detachment during incubation + (10 positives of the 119 strains).

### 3.2. General Parameters and Yeast Cell Count

Table 1 and Table 2 show the results of the general parameters and the yeast cell count (total and viable) of the wines obtained at 3 bars of pressure, respectively.

### 3.3. Comparison of the Proteomic Profiles of Two Yeast Strains

A proteomic analysis was carried out to identify proteins from two strains of *S. cerevisiae* flor yeast, G1, and N62 from a base wine of the Pedro Ximénez variety, independently, subjected to second fermentation conditions in the bottle (Appendix A: Quantification of proteins). A total of 1053 proteins were identified from the LC-MS/MS analysis and after discarding proteins not found in at least 50% of any of the replicates, 921 valid proteins were finally obtained, of which 767 were common in both strains, 133 proteins were identified only in strain N62 (specific), and 21 (specific) proteins in strain G1. The specific proteins corresponded to a qualitative analysis, Figure 1.

### 3.4. Protein Clustering Analysis

#### 3.4.1. Quantification Patterns at the Point with the Highest Protein Abundance

Clustering proteomics from both strains was performed according to quantification pattern. A total of 574 proteins identified in 100% of all biological replicates are included in five clusters with different quantification patterns (Figure 2). First, the quantification value of the proteins was normalized by z-score transformation and clustered according to its pattern. Cluster 1 (*n* = 160) showed a clear difference in quantification between the two strains in the decline from G1 to N62. Cluster 2 (*n* = 254) was characterized by almost identical quantification patterns between the G1 replicates, but for N62, slightly different quantification peaks were observed between replicates, and between the two strains an increase from G1 to N62 was observed. Cluster 3 (*n* = 63) showed small variations between the G1 replicates but exhibited the same quantification patterns for the N62 replicates, as occurred in cluster 1, also showing a decrease from G1 to N62. Cluster 4 (*n* = 26) did not show very significant quantification patterns as a large variability was found between replicates of both strains. Cluster 5 (*n* = 71) showed great similarity in quantification patterns with cluster 2 when comparing both strains with an increase from G1 to N62.

#### 3.4.2. Protein Interaction Networks

Several protein–protein interaction analyses were performed using the STRING v12.0 database of each cluster to investigate the molecular functions associated with each quantification pattern and the specific proteins.

Figure 3a shows that molecular functions related to oxidoreductase and hydrolase activity predominate. In Figure 3b, the functions with the highest enrichment were oxidoreductase, pyruvate decarboxylase, glucokinase, and fructokinase activities. In Figure 3c, the oxidoreductase activity was more significant. In Figure 3e, the oxidoreductase and misfolded protein binding activities showed the greatest enrichment.

Figure 4 shows the INM created from each cluster with a PPI enrichment *p*-value < 0.05.

INM1 (139 edges; PPI enrichment *p*-value < 1.0 × 10^−16^) showed a large number of proteins related to oxidoreductase activity (red nodes), specifically with protection against oxidative stress, reduction in hyperoxides, the biosynthesis of lysine and ethanol, and, on the other hand, with hydrolase activity (blue nodes); within this, function proteins related to the proteasome were found (blue-gray nodes). The remaining proteins related to hydrolase activity have functions in glycogen degradation, glycosidases, and glycosidases in the cell wall, and others are vacuolar proteases.

In INM2 (622 edges; PPI enrichment *p*-value < 1.0 × 10^−16^), proteins were found that were part of many molecular functions, and most of them were related to each other. The most important functions were pyruvate decarboxylase activity (orange nodes), key enzymes in alcoholic fermentation, carbohydrate kinase activity (purple nodes), protein folding chaperone (pink nodes), lyase activity (light gray nodes, some of which are involved in glycolysis), vitamin binding (dark green nodes), ligase activity (light green nodes), oxidoreductase activity (red nodes, some of which are involved in the oxidation of ethanol, and others in protection against oxidative damage by hydrogen peroxide), ATP hydrolysis activity (dark gray nodes), translation regulatory activity (dark burgundy nodes), nucleotide binding (greenish nodes), ribonucleotides binding (light burgundy nodes), mRNA binding (ochre nodes), and hydrolase activity (blue nodes, some of which are vacuolar aminopeptidases, others mitochondrial aminopeptidases, and others metalloproteases).

INM3 (27 edges; PPI enrichment *p*-value < 0.00604) presented proteins associated with ligase activity (green nodes), specifically with carbon–nitrogen bonds for glutamate synthesis, with hydrolase activity (blue nodes), belonging to the peptidase family, and also with oxidoreductase activity (red nodes); some of these proteins are involved in adaptation to oxidative stress, tolerance to aldehydes, and the oxidation of NADPH to NADP^+^, and finally, through catalytic activity (yellow nodes), some of these proteins are involved in the maintenance of the cell wall and the degradation of proteins in the vacuoles.

INM4 (19 edges; PPI enrichment *p*-value < 1.07 × 10^−5^) showed proteins related to hydrolase activity (blue nodes), specifically metallopeptidase, and catalytic activity (yellow nodes) related to the pathway non-oxidative pentose phosphate and glutamate pathway.

INM5 (70 edges; PPI enrichment *p*-value < 5.98 × 10^−5^) showed groups of proteins that are part of the chaperone functions of protein folding (pink nodes), vitamin binding (dark green nodes), and oxidoreductase activity (red nodes) involved in formaldehyde detoxification, the formation of long chain and complex alcohols, the synthesis of ergosterol, and responses to various types of stress, as well as RNA binding (light-greenish nodes); some of these are part of the ribosomal protein complex, while others help to control the length of the poly(A) tail and hydrolase activity (blue nodes); some regulate apoptosis after treatment with H_2_O_2_, while others improve the rapid exchange of oxygen from Pi to water.

The same analysis was also performed for the specific proteins of each strain. In the case of G1 proteins (21), no interaction network was constructed because a PPI enrichment *p*-value < 0.05 was not reached.

The structural component of the ribosome was the most significant and enriched molecular function of the remainder, as shown in Figure 5a. In the case of the specific proteins for N62, an interaction network could be built as shown in Figure 5b in the INM (837 edges; PPI enrichment *p*-value < 1.0 × 10^−16^), which shows a high number of proteins related to the structural component of the ribosome (red nodes), rRNA binding (blue nodes), and translation regulatory activity (green nodes), with some involved in translation initiation or start codon identification, mRNA binding (pink nodes) and nucleic acid binding (yellow nodes). Some of them are essential for the processing and maturation of 27S pre-rRNA and the biogenesis of the large ribosomal subunit, another necessary part for chromatin assembly and chromosome function.

#### 3.4.3. Differential Expression Analysis: ANOVA and HSD Tukey Test

Finally, out of a total of 574 proteins, 127 proteins exceeded the statistical threshold and showed statistical differences in quantification values according to the one-way ANOVA and HSD Tukey tests in comparisons between G1 and N62, with a *q*-value < 0.05. After statistical screening, 63 proteins with significant upward differences were obtained for strain N62, and 64 proteins with significant upward differences were obtained for strain G1, as shown in Figure 6a. These proteins were subjected to another ANOVA and Tukey test, but with a *q*-value < 0.01 and an FC > 2. This resulted in 51 proteins showing a greater quantitative difference. These are shown in a radar plot in Figure 6b.

Of these 51 proteins with high significant differences, 29 proteins showed maximum quantification peaks in strain N62, and 22 proteins presented maximum quantification values in strain G1.

Of the 29 proteins from N62, three are part of cluster 5, and their molecular functions include hydrolase activity, while MAP1 belongs to the peptidase family; for RNA binding, RPS15, both proteins interact with each other as shown in INM5, and the latter is part of the ribosome structure; in the binding of RNAm, PAB1 has an RNA binding function, as observed in INM5, involved in the control of the length of the poly(A) tail. The rest of the proteins with maximum quantification peaks in strain N62 belong to cluster 2, and their molecular functions include lyase activity; one of them is a glyoxalase HSP31, involved in resistance to oxidative stress, and the others are located in the mitochondria. They also interacted with the rest of the proteins of the same cluster, as observed in INM2; oxidoreductase activity, GLR1, has a role in hyperoxia resistance, OYE3 modulates oxidative stress-induced programmed cell death in yeast, and MAE1 catalyzes the oxidative decarboxylation of malate to pyruvate, a key intermediate in sugar metabolism and a precursor for the synthesis of several amino acids. This protein interacts with MDH3; as observed in INM2, it is involved in the glyoxylate cycle and catalyzes the interconversion of malate and oxaloacetate, and CTT1 is a catalase that is responsible for the degradation of hydrogen peroxide, protecting yeast from oxidative stress; hydrolase activity, GPP2 is involved in glycerol biosynthesis; ATP hydrolysis activity, one of which, RPT5, is involved in the degradation of ubiquitinated substrates; SSA4 is important in the orientation and membrane translocation of co-translational proteins and also has significant interaction with several cluster 2 proteins; transferase activity, for QRI1, catalyzes the formation of UDP-N-acetylglucosamine, which is important for cell wall biosynthesis; in kinase activity, POS5 phosphorylates NADH and also interacts with the previous function; in DNA binding, HTB1 is required for chromatin assembly and chromosome function; translation elongation factor, TEF2, binds and delivers amino-acylated tRNA to the A site of ribosomes for the elongation of new polypeptides and has extensive interaction with several proteins as shown in INM2; in rRNA binding, the RPL25 primary ribosomal rRNA-binding protein component of the large ribosomal subunit is involved; the protein folding chaperone SSE1 serves as a nucleotide exchange factor for ATP loading and interacts with a large number of cluster 2 proteins, and SIS1 is involved in the proteasomal degradation of misfolded cytosolic proteins; a structural component of the RPS20 ribosome has many interactions with proteins from the same cluster as observed in INM2; in mRNA binding, NPL3 promotes elongation, regulates termination, and transports poly(A) mRNA from the nucleus to the cytoplasm; in lipid binding, PIL1 is a key protein that regulates mitophagy, autophagy, and cell death, in addition to being essential for the degradation of damaged mitochondria and massive autophagy, protecting against oxidative stress.

Of the 21 proteins that exhibited maximum quantification peaks in strain G1, three are part of cluster 3, and their molecular functions are as follows: in oxidoreductase activity, CCP1 degrades reactive oxygen species in mitochondria, and, involved in the response to oxidative stress, ARI1 uses aromatic and allophatic aldehyde substrates, and also has interactions with other proteins as shown in INM3; in hydrolase activity, EXG1 is involved in the assembly of beta-glucan in the cell wall. The rest of the proteins with maximum quantification values in strain G1 belong to cluster 1, and their molecular functions are as follows: in kinase activity, ADO1 may be involved in the recycling of adenosine produced by the methyl cycle; ERG8 is an essential cytosolic enzyme involved in the biosynthesis of isoprenoids and sterols, including ergosterol; in DNA binding, RFA2 participates in DNA replication, repair, and recombination; a structural component of the ribosome, RPL8B, is also involved; in binding to ubiquitin, EDE1 is a key protein in yeast, as it is the crucial organizer of the initial phase of endocytosis and in the localization of endocytic proteins; in hydrolase activity, UTH1 is located in the inner membrane of mitochondria, and is involved in cell wall biogenesis, the response to oxidative stress, lifespan during starvation, and cell death; PLB1 is involved in the metabolism of lipids, as observed in INM1, where it had interactions with proteins of this cluster; in oxidoreductase activity, LYS9 is in the lysine biosynthetic pathway and has interactions with several proteins as shown in INM1; in GTPase activity, MOG1 and YRB1 are involved in nuclear proteins import, but the latter is also involved in RNA export and ubiquitin-mediated protein degradation during the cell cycle; a structural component of the cell wall, PST1 is secreted by regenerating protoplasts, and is responsible for cell wall maintenance and repair; PIR3 is necessary for cell wall stability; CIS3 is a mannose-containing glycoprotein component of the cell wall; YGP1 is another secretory glycoprotein associated with the cell wall; and the FLO11 protein, which plays a crucial role in cell–cell adhesion and biofilm formation under conditions of nutrient limitation, is also involved.

## 4. Discussion

In this first approximation in the conditions tested, a comparative proteomic study of the two strains analyzed would reveal the main metabolic pathways in terms of carbon and energy flow to be glycolysis and alcoholic fermentation. The SGA1 protein, which is involved in glycogen degradation, was significantly found in strain G1, which could be due to the fact that this strain would have a greater reserve of this polysaccharide [17]. In terms of glycolysis, the significantly more abundant proteins in strain N62 were HXK1, HXK2, ENO1, and ENO2 to form pyruvate, so this strain could be obtaining more energy and compounds for biomass production (Figure 7). This could be related to the higher growth and viability found in this strain. On the other hand, a significant TPI1 protein was obtained in G1, which, in addition to being involved in the pathway for glycerol biosynthesis, and in obtaining NAD^+^ [18], it may also be related to the recycling of glucose-6-phosphate through the non-oxidative pentose phosphate pathway caused by the limitation of glycolytic flux [19].

In G1, the significantly different ADH1 protein involved in the reduction of acetaldehyde to ethanol was found. This may be related to the higher ethanol concentration achieved by this strain (see Table 1). In strain N62, a protein from the dehydrogenase family but with a different function, SFA1, was also found to be significant, as can be seen in IMN5. High levels of SFA1 have been shown to support formaldehyde tolerance [20,21]. ADH4 is a specific protein of the N62 strain. It is a poorly understood protein that localizes to the mitochondria. It does not appear to produce ethanol, although it may be expressed during fermentation when ethanol production is high [22]. In N62, the protein PDC6, which is responsible for the decarboxylation of pyruvate to acetaldehyde, and ALD4, which converts acetaldehyde to acetate in the mitochondria, were obtained [23,24]. This is also consistent with the higher production of acetic acid by this strain (see Table 1), where volatile acidity is approximately twice as high in N62 as in G1.

After ethanol and carbon dioxide, the quantitatively most important fermentation product is glycerol, the amount of which depends on several environmental factors and the yeast strain [25]. This compound helps to improve the sensory quality of the wine by providing viscosity and smoothness [7]. It also helps to cope with osmotic stress, control the redox balance, and convert NADH to NAD^+^ [26]. In N62, the DAK1 and GPP1 proteins were significant. These are related to the glycerol biosynthesis pathway, which could explain why the concentration of this polyol was higher in wines made with the N62 strain than with the G1 strain (see Table 1). The GPP1 protein is induced by anaerobic and osmotic stresses [27], and GPP2 is induced in response to hyperosmotic and oxidative stresses [13].

Regarding oxidative stress, the related protein in N62 is HSP31, which also responds to osmotic and thermal stresses; its concentration is higher the more ethanol is produced during fermentation [28]. Also, according to INM2 and Natkańska et al. [29], it has interaction and could be associated with chaperones, specifically with HSP78, which has shown significant differences and can prevent the aggregation of misfolded proteins. This in turn interacts with other chaperones, SSE1, SSE2, and another related stress-induced protein, SSA4. These three proteins belong to the same HSP70 family, so they participate in binding to ATP. The POS5 protein is considered a great source of mitochondrial NADPH and has a better NADH kinase activity than NAD^+^ kinase activity, which may indicate a great synthesis of NADPH from NADH with consumed ATP, both abundant in the mitochondria of this strain [30]. On the other hand, the regeneration of NADPH from NADP^+^ can be carried out by ALD4 and MAE1 [31], which are proteins that show interaction with POS5 and, like this, present significant differences with respect to the other strain. Therefore, it could be said that all these reactions are more active in N62 than in G1. CTT1 participates in oxidative stress by being responsible for breaking down intracellular hydrogen peroxide that is formed from the synthesis of ATP during respiratory metabolism [32,33]. In INM2, it is shown that it interacts with ALD4, and this may be because acetate can convert to acetyl-CoA and enter the TCA, as with GLR1, it also has an interaction. This protein is responsible for transforming oxidized glutathione into its reduced form. A decrease in GLR1 can slow down cell growth, since together with NADPH it is one of the best defense systems against oxidants [34]. Finally, the OYE3 protein could reduce reactive oxygen species and increase resistance to programmed cell death induced by hydrogen peroxide [35]. Due to the oxidative stress that occurs under these study conditions, the RNY1 protein was identified with significant differences, which is responsible for separating tRNAs, and they are exported from the vacuoles to the cytosol, promoting cell death [36].

In the G1 strain, the proteins with significant differences that were obtained include the CPP1 protein; it seems that it can act more as a hydrogen peroxide mitochondrial sensor than as a scavenger of this compound [37]. The proteins TRX2 and TRX3, involved in oxidative and reductive stress, also showed significant values, the first in the cytosol and the second in the mitochondria. They carry out disulfide exchange reactions on oxidized substrates, including ribonucleotide reductase and numerous oxidative defense enzymes [38].

In strain G1, a large number of proteins significantly different from the cell wall were quantified, which is formed by a network of polysaccharides that are subject to changes allowing the cell to grow or/and face cell lysis [39]. An increase in YGP1, asecretory glycoprotein related to cell wall biogenesis [40], can reduce ethanol production and mitochondrial activity in addition to favoring the expression of EXG1, SCW11, PST1, CIS3, and PIR3, and impairs the expression of QRI1 [41], as our results have revealed, based on the quantification of YGP1. SMI1 is involved in cell wall integrity, synthesis, and stress response [42]. A higher quantification of EXG1 protein may also be due to the SMI1 protein, which may increase its intensity [43]. EXG1 is one of the main glucanases of the cell wall necessary for its maintenance, the integration of mannoproteins in this structure, and the binding of beta-glucan [35]. The CIS3 and PIR3 proteins belong to the same family of mannoproteins. Both carry out processes and functions such as synthesis and cell wall stability, in addition to being related to thermal and nutritional stress [44]; in addition, they are known to be important for survival in the stationary growth phase [45]. Another identified mannoprotein, PST1, is important for the integrity and response to damage to the cell wall [46]. A higher mannoprotein concentration is considered to make an improvement in wine quality and organoleptic properties [47]. This high concentration may also be associated with a better response to stress [48]. EMC33 is important for the architecture and integrity of the cell wall, as well as for the proper binding of the outer layer of mannoproteins [49]. The flocculation capacity of the selected yeasts is a convenient factor to produce sparkling wines, because it allows for the rapid clarification of the fermenting product, thereby reducing production time and cost [50]. The main protein of this cell adhesion is the floculin FLO11, found in both strains, expressed significantly and in higher abundance in the G1 strain [51,52]. The presence of this protein in both strains could be explained by the observation of flocs in the bottles. UTH1 can reduce the concentration of some mitochondrial proteins, so it may have the ability to regulate mitochondrial function. It also appears that it may affect the function of the yeast cell wall [53]. NCA3 and UTH1 belong to the same family of SUN proteins, and these proteins participate in mitochondrial biogenesis, autophagy, cytokinesis, cell wall structure, and DNA replication [53]. These proteins are involved in reforming the cell wall during the transition from the exponential to stationary phase [54]. In strain N62, a protein with significant differences related to the cell wall was identified, QRI1, which synthesizes the chitin found in the cell wall [55]. This protein was quantified at a higher level because YPG1 was expressed at a lower level in this strain according to Arnthong et al. [41].

In strain G1, two proteins related to endocytosis have been quantified with significant differences. EDE1 and CLC1 are the first proteins to reach endocytic sites [56]. Furthermore, they are important for good dynamics in endocytic proteins, which arrive afterwards and for cargo adsorption [57]. Most ergosterol in fungi is located in the plasma membrane [58], as well as in the vacuole membranes [59]. This sterol is important for good endocytic apparatus execution [60]. This could explain why ERG8, which is part of ergosterol biosynthesis [61], presents significant differences in this strain. In the N62 strain, the PIL1 protein could inhibit protein kinases that participate in signaling pathways for cell wall integrity [62] and participate in endocytosis [63]. The specific proteins of this strain identified SEC18, which is necessary for vesicle-mediated transport [64], EIS1, to form eisosomes [65], which can respond to different types of stress [66], and LSP1. This, together with PIL1, is an essential protein in the formation of eisosomes, thus allowing for the endocytosis process [60]. It has been discovered that PIL1 is also found in the outer membrane of mitochondria, and in its absence, mitophagy occurs [67]; this could explain the differences in quantification between strains, since in G1 this protein has been identified in a lower proportion, and this might indicate that mitophagy has begun and initiates the autophagy process. G1 reached 3 bars of pressure at 52 days and the N62 strain at 27 days. These results could explain why in G1 proteases have been identified with significant differences related to their degradation in the vacuoles, signs that indicate the beginning of autophagy, such as proteins PRC1, PRB1, ATG42/YBR139W, and proteasome-related proteins, PRE8, PRE9, and PUP2, which also participate in autophagy [68,69].

Finally, in the N62 strain, a greater number of proteins were obtained with significant differences and specificities related to ribosomes and the transcription and translation of proteins [70]. The protein involved in transcription in this strain was TEF2, and specifically, the following: RSC6, SPT5, NCL1, and ARP5. Those in translation were the following: PAB1, GCD11, MAP1, and GIS2. Specifically, SUP35, PRT1, SUI3, SUP45, RPG1, DED81, FUN12, TPA1, and MKT1 were found in protein synthesis or part of the ribosome, as well as RPL25, RPS20, RPS15, SSZ1, and RPS14B; proteins from part of the mitochondrial ribosome include MRPL3 and, specifically, RPL28, RPL7A, RPL33A, RPL22A, RPS13, RPL27A, RPL31A, RPL20B, RPS24A, RPS4A, RPS6B, RPL23A, RPS11B, RPL4A, RPS17B, RPL16B, NHP2, MRT4, RPS1A, RPL14B, RPL34B, RPL10, RPS10B, RPL4B, RPL6A, RPL21A, EMG1, FAL1, RPS22B, ZUO1, and NOP2. Also specifically associated with mitochondrial ribosomes were the following: MRP20, MRPS5, MRPL7, MRPL40, MRPS9, MRPL49, MRPS18, RSM26, YML6, MRPS12, RSM23, MRPS8, MRLP1, MRPL51, and MRPL35. Those related to mRNA were as follows: NPL3 and PUB1. Specifically, this involved NAM7, PRP19, NAB3, PUF6, PUS7, and PUS1. All of this is closely associated with the high rate of cell growth and viability observed in this strain.

## 5. Conclusions

A comprehensive comparative analysis of the proteomic profiles of two indigenous *S. cerevisiae* yeast strains isolated from fine wine veils midway in the second fermentation in the bottle has been carried out. Important qualitative differences and significant quantitative differences, some of which could be key to explain cell growth and viability, were found in the production of glycerol, ethanol, and acetic acid (volatile acidity), and proteins related to cell wall and autophagy. These interesting results may indicate that strain N62 may be excellent for producing higher-quality sparkling wines. This study can contribute to the wider field of oenology by providing detailed proteomic data on indigenous yeast strains, which could lead to innovations in sparkling wine production techniques and the development of new flavor profiles in sparkling wines, as well as in finding protein biomarkers that can be used to select strains for use in sparkling wine production.

## Figures and Tables

**Figure 1 foods-14-00282-f001:**
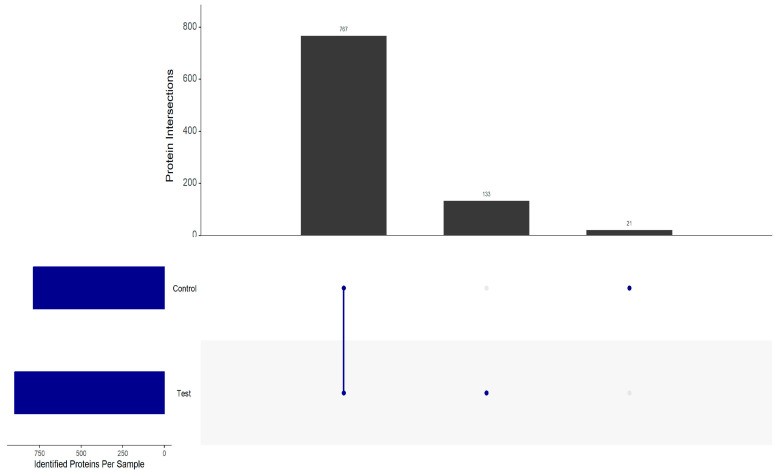
Intersection diagram of the proteins of the two yeast strains identified in at least 50% of the total of some of the samples. The number of proteins in each intersection group is represented by the bars.

**Figure 2 foods-14-00282-f002:**
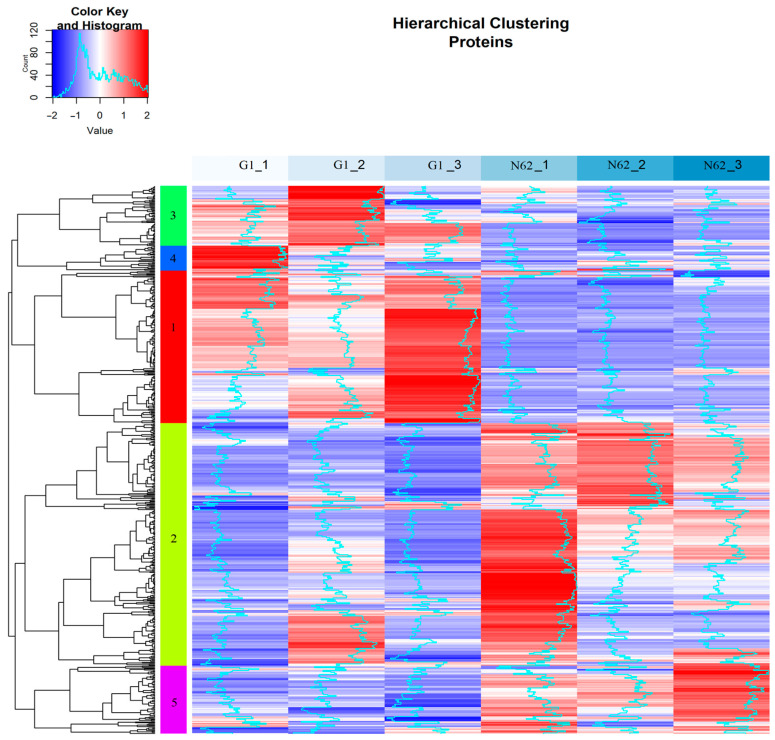
Heat map showing the hierarchical clustering analysis of the proteins identified in 100% of the total samples. The number in each color represents each of the five groups.

**Figure 3 foods-14-00282-f003:**
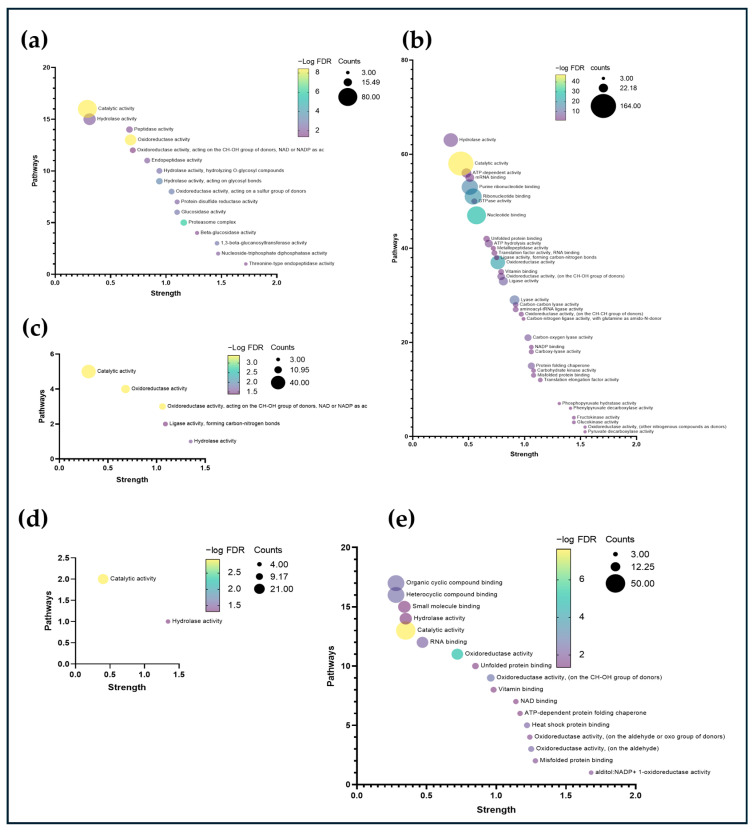
Bubble plot of the enrichment terms of the analysis of the set of proteins according to their quantification pattern (**a**–**e**) using the “Molecular Function” term from the Gene Ontology database. The color gradient refers to the significance of each function (FDR < 0.05). The size of the bubbles represents the number of proteins that correspond to each of the functions.

**Figure 4 foods-14-00282-f004:**
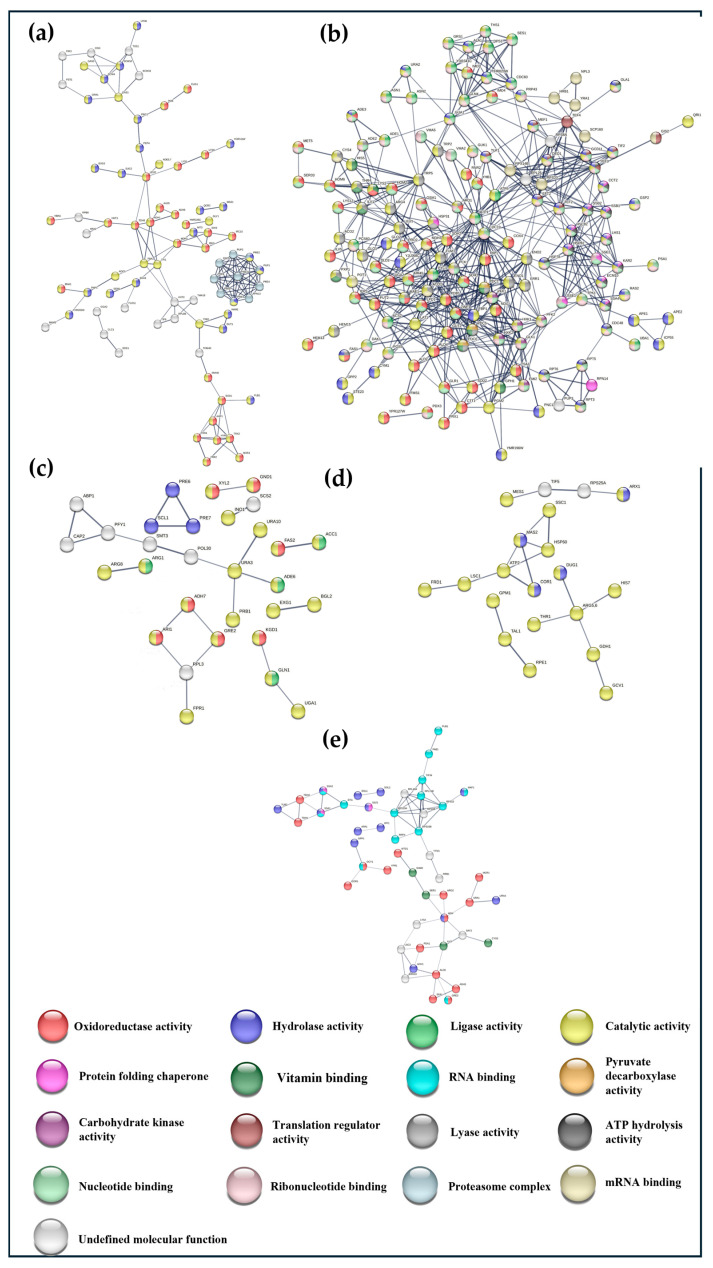
Protein–protein interaction network maps (INM) between proteins with the same quantification patterns (**a**–**e**) constructed with STRING v12.0, high confidence, with an FDR value < 0.05. The proteins are shown as nodes and the interactions between them as edges. Nodes of the same color correspond to the same molecular function according to the Gene Ontology bases.

**Figure 5 foods-14-00282-f005:**
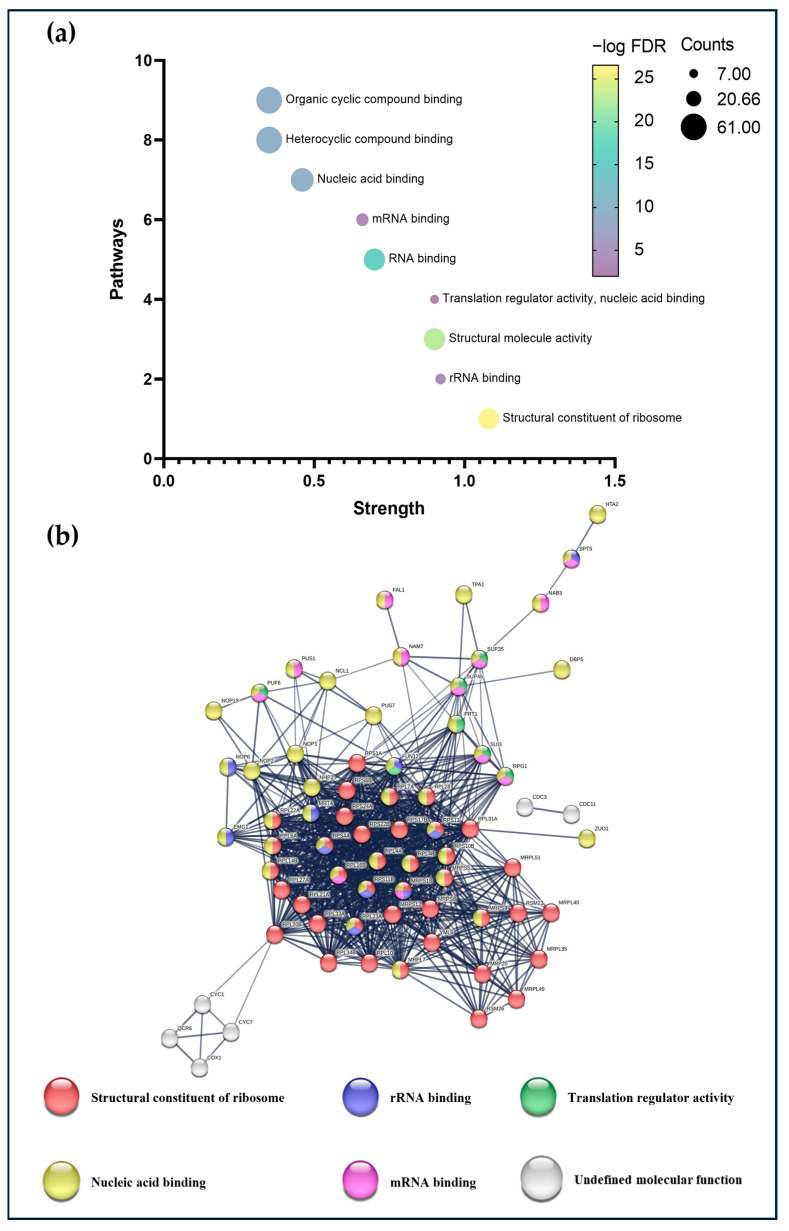
(**a**) Bubble plot and (**b**) protein–protein interaction network map of the N62 strain-specific proteins.

**Figure 6 foods-14-00282-f006:**
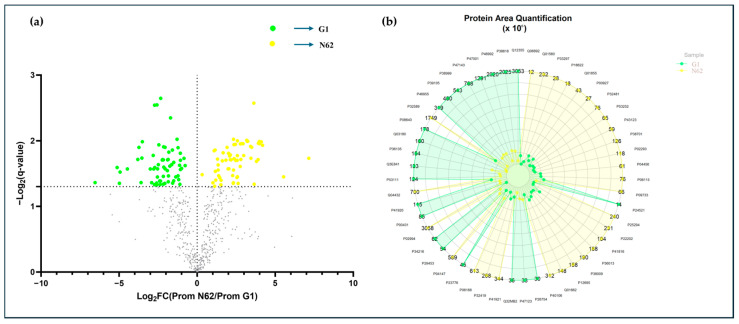
(**a**) Volcano plot of proteins with significant differences at *q*-value < 0.05. The up-regulated proteins correspond to the yellow spheres (LFC > 0) and the down-regulated proteins correspond to the green spheres (LFC < 0). Proteins without significant differences are shown in gray. (**b**) Radial plot of proteins with the strongest significant differences according to the Tukey HSD test corrected for multiple testing (*q*-value < 0.01) and log_2_ fold change (in absolute value) > 2.

**Figure 7 foods-14-00282-f007:**
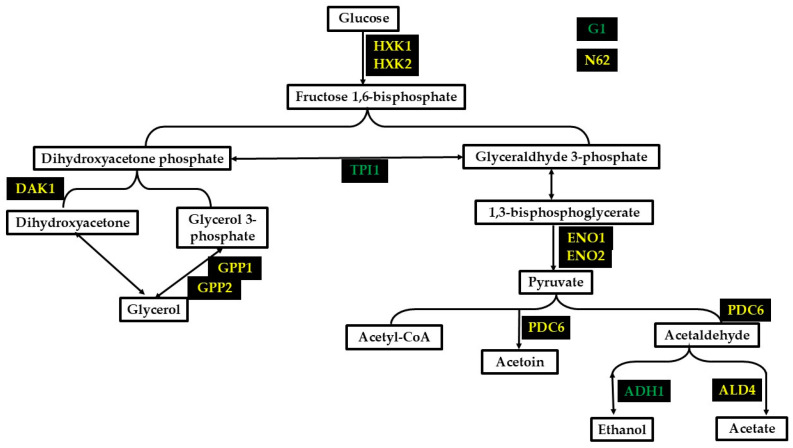
Most notable proteins with significant differences involved in the second fermentation in the bottle at 3 bars of pressure. The color in the name of the proteins represents a greater amount in yellow (N62) and green (G1).

**Table 1 foods-14-00282-t001:** Physicochemical parameters of the wines at 3 bars of pressure (* mean *q* < 0.05; **** mean *q* < 0.0001).

	G1	N62	*q*-Value
pH	3.022 ± 0.071	3.132 ± 0.061	1.079
Reducing sugar (g/L)	0.235 ± 0.004	0.221 ± 0.003	1.308
Volatile acidity (g/L)	0.224 ± 0.021	0.421 ± 0.111	0.048 *
Titratable acidity (g/L)	4.343 ± 0.613	4.260 ± 0.111	0.924
Ethanol (% *v*/*v*)	11.929 ± 0.080	11.100 ± 0.080	7.736 × 10^−8^ ****
Glycerol (mg/L)	5570.777 ± 55.641	6875 ± 55.784	5.529 × 10^−12^ ****

**Table 2 foods-14-00282-t002:** Total and viable cell count of both strains at 3 bars of pressure and time reached. (** mean *q* < 0.01; **** mean *q* < 0.0001).

	G1	N62	*q*-Value
Viable cells/mL	0.730 × 10^6^ ± 0.035	3.900 × 10^6^ ± 0.265	0.005 **
Total cells/mL	21.000 × 10^6^ ± 1.730	46.300 × 10^6^ ± 3.210	1.939 × 10^−5^ ****
Time (days)	52	27	1.016 × 10^−5^ ****

## Data Availability

The original contributions presented in this study are included in the article/Appendix A. Further inquiries can be directed to the corresponding author.

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
