# Peer review of "Comparative Proteomics of Two Flor Yeasts in Sparkling Wine Fermentation: First Approach"

_foods, 2025, doi:10.3390/foods14020282_

Round 1

Reviewer 1 Report

Comments and Suggestions for Authors

General/major comments

In this paper the authors make a comparative proteomic study of two yeasts used for the production of sparkling wine. The authors chose to make this comparison when the pressure in the wine reached 3 atm. Since fermentation is not a static phenomenon, it would have been interesting to perform kinetics and compare protein profiles at different points in the fermentation process. Although this is an important and interesting study, the qualitative information that emerges from it is quite thin. Indeed, the differences in fermentation properties between the two yeast strains are already perceptible if we look at the physicochemical parameters of the wines. My main concern about this work deals with the absence of kinetics, and the authors should better justify their choice of having done a proteomic study rather than a transcriptomic study. I nevertheless suggest accepting this paper for publication after taking into account the minor remarks related to this report.

Specific comments

L68: Has the N62 strain been deposited in a strain collection and if so, please specify.

L206-208: Check this sentence which is not clear, punctuation problem in particular.

L217: Change the title of table 1 which shows the physicochemical parameters of the wines but not the yeasts parameters.

L253: Change “analyze” to “analysis”

L458: Please change the sentence “Table 1, the volatile acidity of strain N62 was almost double that of the other strain” to “Table 1, the volatile acidity obtained with the strain N62 was almost double of that obtained with the other strain”.

L468: Please change “stress” to “stresses”

L468-469: use the singular in the sentence: “the related protein in N62 is HSP31, ….to osmotic and thermal STRESSES, its concentration is higher the more ethanol is produced during fermentation.

L557-558: Please clarify this sentence.

L563-564: This sentence is quite unclear and needs to be changed.

Comments on the Quality of English Language

It would be appropriate to have the paper read by an English speaker to make it more fluid.

Author Response

In this paper the authors make a comparative proteomic study of two yeasts used for the production of sparkling wine. The authors chose to make this comparison when the pressure in the wine reached 3 atm. Since fermentation is not a static phenomenon, it would have been interesting to perform kinetics and compare protein profiles at different points in the fermentation process. Although this is an important and interesting study, the qualitative information that emerges from it is quite thin. Indeed, the differences in fermentation properties between the two yeast strains are already perceptible if we look at the physicochemical parameters of the wines. My main concern about this work deals with the absence of kinetics, and the authors should better justify their choice of having done a proteomic study rather than a transcriptomic study. I nevertheless suggest accepting this paper for publication after taking into account the minor remarks related to this report.

Thank you very much for your comments and suggestions. We chose 3 atmospheres because it is approximately the middle of the fermentation and according to previous work there is a high abundance of proteins at this point, it could also serve as a point of comparison with those works. To make a kinetic, which would be the most correct approach, would mean a greater economic effort, proteomics is still expensive today, more so than transcriptomics, and a greater amount of data would have to be handled, so a lot of work would have to be done. This work is the first approach to the potential use of a new yeast in sparkling wine production. Our research group is an expert in proteomics of wine yeasts, and we decided to carry out this study because proteins are closer to metabolites.

 Specific comments

L68: Has the N62 strain been deposited in a strain collection and if so, please specify.

The place of deposit has been included in the revised manuscript.

L206-208: Check this sentence which is not clear, punctuation problem in particular.

You are right, the paragraph has been changed for clarity.

L217: Change the title of table 1 which shows the physicochemical parameters of the wines but not the yeasts parameters.

The title of Table 1 has been changed.

L253: Change “analyze” to “analysis”

“Analyze” has been changed to “analysis” in the revised manuscript.

L458: Please change the sentence “Table 1, the volatile acidity of strain N62 was almost double that of the other strain” to “Table 1, the volatile acidity obtained with the strain N62 was almost double of that obtained with the other strain”.

The sentence has been changed in the revised manuscript.

L468: Please change “stress” to “stresses”

“Stress” has been changed to “stresses” in the revised manuscript.

L468-469: use the singular in the sentence: “the related protein in N62 is HSP31, ….to osmotic and thermal STRESSES, its concentration is higher the more ethanol is produced during fermentation.

This has been changed in the revised version of the manuscript.

L557-558: Please clarify this sentence.

This paragraph has been simplified, mostly removed, considering the reviewer's suggestions.

L563-564: This sentence is quite unclear and needs to be changed.

This paragraph has been simplified, mostly removed, considering the reviewer's suggestions.

Comments on the Quality of English Language

It would be appropriate to have the paper read by an English speaker to make it more fluid.

The manuscript has been thoroughly revised, and the English has been corrected.

Reviewer 2 Report

Comments and Suggestions for Authors

In this study, the proteomic profiles of yeast strain G1 and N62 were compared. However, the manuscript did not provide corresponding results to support the conclusion. The writing of the manuscript is very confusing.

1.     In Title, the innovation of the manuscript was not reflected.

2.     In Abstract, there were too much background introduction, but too few results.

3.     Line 20-23, “The results showed that these strains behave differently: the strain that reached 3 atmospheres faster was strain N62, which achieved a higher biomass, glycerol content and abundance of proteins related to protein synthesis, and in G1 a higher ethanol content was reached, and the onset of autophagy was detected. Is there any test result to support the difference in the biomass, glycerol content, proteins, and ethanol content between two strains? The connection between these differences and proteomic analysis should be supplemented.

4.     In Introduction, research on yeast proteomics needs to be supplemented.

5.     Line 134, “Tris-HCL” should be changed to “Tris-HCl”.

6.     Line 211-212, the screening results of strain N62 need to be supplemented.

7.     3.3. Protein clustering analysis, more information on the differences between the two strains needs to be supplemented, while irrelevant information needs to be extensively deleted.

8.     Line 581-583, is there any test result to support that strain N62 may be excellent to produce higher quality sparkling wines, due to its positive influence on the structure and body of the wine, as well as specific aromatic notes?

Author Response

Comments and Suggestions for Authors

In this study, the proteomic profiles of yeast strain G1 and N62 were compared. However, the manuscript did not provide corresponding results to support the conclusion. The writing of the manuscript is very confusing.

Dear reviewer, thank you for your comments. The suggestions have been considered and will undoubtedly improve the quality of the manuscript.

  1. In Title, the innovation of the manuscript was not reflected.

The innovation of this article lies in the use of native flor yeast strains, typical of the biological ageing of Sherry wines, for the production of sparkling wines. A comparative study has even been carried out from the proteomic point of view, which is the level of expression closest to the final product in terms of its organoleptic characteristics.

Following your suggestion we have changed the title to: Comparative Proteomics of Two Flor Yeasts in Sparkling Wine Fermentation: A First Approach.

  1. In Abstract, there were too much background introduction, but too few results.

We have removed some of the background and included relevant results according to your suggestion.

  1. Line 20-23, “The results showed that these strains behave differently: the strain that reached 3 atmospheres faster was strain N62, which achieved a higher biomass, glycerol content and abundance of proteins related to protein synthesis, and in G1 a higher ethanol content was reached, and the onset of autophagy was detected“. Is there any test result to support the difference in the biomass, glycerol content, proteins, and ethanol content between two strains? The connection between these differences and proteomic analysis should be supplemented.

The relationship between these differences and the proteomic analysis has been added at the end of the abstract.

  1. In Introduction, research on yeast proteomics needs to be supplemented.

New research on yeast proteomics in sparkling wine has been included in the revised manuscript.

  1. Line 134, “Tris-HCL” should be changed to “Tris-HCl”.

Corrected.

  1. Line 211-212, the screening results of strain N62 need to be supplemented.

The screening results of strain N62 have been completed.

  1. 3.3. Protein clustering analysis, more information on the differences between the two strains needs to be supplemented, while irrelevant information needs to be extensively deleted.

This section has been thoroughly revised for clarity.

  1. Line 581-583, is there any test result to support that strain N62 may be excellent to produce higher quality sparkling wines, due to its positive influence on the structure and body of the wine, as well as specific aromatic notes?

It is known from the literature that wines with a high concentration of glycerol have a positive effect on the quality of these wines. As N62 produces a higher amount of glycerol, it is expected that the finished wines will have more body and structure. Glycerol contributes to the wine's texture and mouthfeel, but not aromatic notes (doi:10.1006/fstl.2001.0766). This has been clarified in the revised manuscript.

Reviewer 3 Report

Comments and Suggestions for Authors

Abstract: please add key data to support your conclusion.

Keywords: Saccharomyces cerevisiae; proteins; sparkling wineï¼›second fermentation

Line 130, convert rpm into g, please revise the same mistakes in the manuscript.

Line 152-154, the sentence is unreadable, please revise it.

Comments on the Quality of English Language

bstract: please add key data to support your conclusion.

Keywords: Saccharomyces cerevisiae; proteins; sparkling wineï¼›second fermentation

Line 130, convert rpm into g, please revise the same mistakes in the manuscript.

Line 152-154, the sentence is unreadable, please revise it.

Author Response

Comments and Suggestions for Authors

Dear reviewer, thank you for your comments and suggestions.

Abstract: please add key data to support your conclusion.

Data have been added

Keywords: Saccharomyces cerevisiae; proteins; sparkling wineï¼›second fermentation

Corrected.

Line 130, convert rpm into g, please revise the same mistakes in the manuscript.

rpm has been converted to g throughout the manuscript.

Line 152-154, the sentence is unreadable, please revise it.

Corrected.

Round 2

Reviewer 2 Report

Comments and Suggestions for Authors

The manuscript has been significantly revised and can be accepted.

Author Response

Thank you very much for your comments and effort on this manuscript.

Reviewer 3 Report

Comments and Suggestions for Authors

Table 1 and 2, please provide the significant differences between the numerical values.

Figures, please enhance the resolution of figures.

Author Response

Table 1 and 2, please provide the significant differences between the numerical values.

The significance of the results is shown in Tables 1 and 2.

Figures, please enhance the resolution of figures.

The resolution of the figures has been improved.